# Farnesol-Loaded Liposomes Protect the Epidermis and Dermis from PM_2.5_-Induced Cutaneous Injury

**DOI:** 10.3390/ijms22116076

**Published:** 2021-06-04

**Authors:** Yu-Chiuan Wu, Wei-Yun Chen, Chun-Yin Chen, Sheng I. Lee, Yu-Wen Wang, Han-Hsiang Huang, Shyh-Ming Kuo

**Affiliations:** 1Hualien Armed Forces General Hospital, Hualien County 97144, Taiwan; ranger.wu1113@gmail.com (Y.-C.W.); vivian07280603@gmail.com (W.-Y.C.); 2School of Culinary Arts, National Kaohsiung University of Hospitality and Tourism, Kaohsiung City 81271, Taiwan; 3Department of Biomedical Engineering, I-Shou University, Kaohsiung City 84001, Taiwan; memtal0825@gmail.com (C.-Y.C.); kittysh981026@gmail.com (S.I.L.); yuwen870928@gmail.com (Y.-W.W.); 4Department of Veterinary Medicine, National Chiayi University, Chiayi City 60054, Taiwan

**Keywords:** PM_2.5_ particle, farnesol, liposome, injuries and inflammation, therapeutic and protective effects

## Abstract

Particulate matter with aerodynamic diameter ≤2.5 μm (PM_2.5_) increases oxidative stress through free radical generation and incomplete volatilization. In addition to affecting the respiratory system, PM_2.5_ causes aging- and inflammation-related damage to skin. Farnesol (Farn), a natural benzyl semiterpene, possesses anti-inflammatory, antioxidative, and antibacterial properties. However, because of its poor water solubility and cytotoxicity at high concentrations, the biomedical applications of Farn have been limited. This study examined the deleterious effects of PM_2.5_ on the epidermis and dermis. In addition, Farn-encapsulated liposomes (Lipo-Farn) and gelatin/HA/xanthan gel containing Lipo-Farn were prepared and applied in vivo to repair and alleviate PM_2.5_-induced damage and inflammation in skin. The prepared Lipo-Farn was 342 ± 90 nm in diameter with an encapsulation rate of 69%; the encapsulation significantly reduced the cytotoxicity of Farn. Lipo-Farn exhibited a slow-release rate of 35% after 192 h of incubation. The half-maximal inhibitory concentration of PM_2.5_ was approximately 850 μg/mL, and ≥400 μg/mL PM_2.5_ significantly increased IL-6 production in skin fibroblasts. Severe impairment in the epidermis and hair follicles and moderate impairment in the dermis were found in the groups treated with post-PM_2.5_ and continuous subcutaneous injection of PM_2.5_. Acute and chronic inflammation was observed in the skin in both experimental categories in vivo. Treatment with 4 mM Lipo-Farn largely repaired PM_2.5_-induced injury in the epidermis and dermis, restored injured hair follicles, and alleviated acute and chronic inflammation induced by PM_2.5_ in rat skin. In addition, treatment with 4 mM pure Farn and 2 mM Lipo-Farn exerted moderate reparative and anti-inflammatory effects on impaired skin. The findings of the current study indicate the therapeutic and protective effects of Lipo-Farn against various injuries caused by PM_2.5_ in the pilosebaceous units, epidermis, and dermis of skin.

## 1. Introduction

Exposure to particulate matter and ozone severely affects public health worldwide. In recent years, fine particulate matter (or particulate matter with aerodynamic diameter ≤2.5 μm (PM_2.5_)) was reported to exert harmful effects on several major systems and organs including the respiratory, cardiovascular, and immune systems. Studies have demonstrated an association of PM_2.5_ with aggravation of asthma and chronic obstructive pulmonary disease (COPD), death due to cardiovascular disease, and lung cancer [1,2,3,4]. Among the systems and organs affected by PM_2.5_, the skin is the external natural barrier and primary organ exposed to ambient pollutants and protects the human body by acting as an interface between the body and surrounding atmosphere. The effects of PM_2.5_ on human skin have not been fully clarified. Many studies have indicated that air pollution may damage and exert deleterious effects on skin; for instance, an increase in the oxidative stress in skin can aggravate atopic dermatitis. Oxidative stress is a common mechanism underlying PM_2.5_-induced skin damage. The possible adverse effects of PM_2.5_ on skin have attracted the attention of clinical dermatologists and researchers. PM_2.5_ could induce oxidative stress by increasing the number of reactive oxygen species, leading to DNA damage, lipid peroxidation, and protein carbonylation; these factors further resulted in endoplasmic reticulum stress, mitochondrial dysfunction, and autophagy, thus causing apoptotic HaCaT cell death and mouse skin injury [5]. Similarly, inflammatory responses and oxidative stress caused by PM_0.3–2.5_ in an in vitro model negatively affected the integrity and functions of skin [6]. PM_2.5_ can penetrate both disrupted and intact skin; it disrupts the skin barrier by reducing the numbers of cytokeratin, filaggrin, E-cadherin, and tight junction molecules [7,8]. In addition, studies on the respiratory system have indicated that PM_2.5_ inhalation increased the levels of proinflammatory cytokines such as interleukin (IL)-1β, IL-6, and tumor necrosis factor-α (TNF-α), demonstrating that PM_2.5_ is strongly correlated with inflammatory responses in the airway and in patients with COPD [9,10]. Moreover, exposure of the circulatory system to PM_2.5_ can contribute to atherogenesis and acute cardiovascular and cerebrovascular events [11]. Similarly, an in vitro study reported that the production of IL-6, TNF-α, and matrix metalloproteinase-1 in a keratinocyte and skin fibroblast coculture system was largely increased, whereas that of procollagen type I was significantly decreased after exposure to PM_2.5_ [12], indicating that PM_2.5_ can negatively affect skin cells and lead to inflammatory responses in the epidermis and dermis. Most of the PM_2.5_-associated events that contribute to cellular dysfunction, inflammation, and extracellular matrix destruction may exert severe deleterious effects on skin. 

Farnesol (Farn), a natural 15-carbon organic compound, possesses many biological and biomedical properties [13,14]. Farn can exert numerous therapeutic effects including antimicrobial, antiproliferative, antiallergic, and anti-inflammatory effects as well as induce apoptosis [15,16,17]. In addition, studies have examined the effects of drugs or potential therapeutic compounds encapsulated by biocompatible and biodegradable liposomes. Liposomes can entrap both hydrophobic and hydrophilic drugs. Our previous studies have indicated that liposomal encapsulation could prolong and reinforce the effects of drugs by protecting them from degradation and could reduce the drug side effects [14,18,19]. Furthermore, our previous studies have revealed the novel therapeutic effects of different Farn dosages on skin. Liposomal or pure Farn protected against ultraviolet B (UVB), increased collagen production, and improved skin smoothness while exerting anti-inflammatory and tissue-reparative effects on UVB-caused sunburn and third-degree burns in the epidermis in vitro and in the dermis in vivo. Farnesol has been found to promote wound healing by suppressing the production of proinflammatory cytokines such as interleukin (IL)-1 beta, IL-6, and tumor necrosis factor (TNF)-alpha [14,20,21]. The present study investigated the deleterious effects of PM_2.5_ on the epidermis and dermis and examined the anti-inflammatory and restoration-promoting effects of pure and liposomal Farn on various skin injuries caused by PM_2.5_.

## 2. Results

### 2.1. Size Measurement, Entrapment Efficiency, and In Vitro Release of Lipo-Farn

The size of the resultant Lipo-Farn was calculated on a random sampling basis for approximately 100–150 individual liposomes through transmission electron microscopy. The average size of Lipo-Farn in the current study was 342 ± 90 nm (Figure 1A). The Entrapment efficiency (EE) of Lipo-Farn was measured by the formula described in Section 4.2. As a result, an EE of 69% for the fabricated Lipo-Farn in this study was obtained (*n* = 3). The in vitro release profile of Lipo-Farn was further examined. After 24 h of incubation, approximately 26% farnesol was released from Lipo-Farn. The slow in vitro release profile of Lipo-Farn around 35% was acquired after 192 h of incubation (Figure 1B).

### 2.2. Effects of PM_2.5_, Farn and Lipo-Farn on Cell Viability, Morphology and IL-6 Production In Vitro

The in vitro effects of PM_2.5_ on skin fibroblast viability and morphology were firstly analyzed. The cell viability of skin fibroblasts was significantly inhibited after treatment with ≥800 μg/mL PM_2.5_, while exposure to 600 μg/mL PM_2.5_ showed >20% inhibition on skin fibroblast viability (Figure 2A). The skin fibroblasts had lost their normal spindle-shaped features after treatment with ≥200 μg/mL PM_2.5_ (Figure 2C). Moreover, the skin fibroblasts produced a significantly increased level of IL-6 after treatment with ≥400 μg/mL PM_2.5_ (Figure 2B). The inhibition of the viability of skin fibroblasts by pure Farn was found to be >20% at a concentration of >0.2mM (Figure 3A), as treatment with 0.4 mM pure Farn caused loss of the normal spindle-like shape in around 50% of skin fibroblasts (Figure 3B). Fluorescence staining of live cells represented nearly identical results with those assessed by MTT assay, in which live cells were <40% when the concentrations of pure Farn over 0.4 mM (Figure 3C). Cell viability of skin fibroblasts was not significantly inhibited by Lipo-Farn from 0.1 to 1.0 mM (Figure 4A). The skin fibroblasts exposed to 0.1 mM to 1.0 mM Lipo-Farn barely showed apparent changes in cell morphology (Figure 4B), but fluorescence staining showed that the decreases in live cells was approximately 20% after treatment with 1.0 mM Lipo-Farn (Figure 4C).

### 2.3. In Vivo Histopathological Results and Wound Healing Scores

In vivo histopathological analysis showed that PM_2.5_ exerted adverse effects on the epidermis in the untreated groups of both categories (Figure 5 and Figure 6). Numerous epithelial and follicle cysts arising from injured pilosebaceous units were surrounded by inspissated keratin in the epidermis. Moreover, rupture of the cyst lining and loss of keratin were found in the group that received no treatment after PM_2.5_ exposure (Figure 5) and the group that received continuous exposure of PM_2.5_ without treatment (Figure 6). Epidermal erosion was observed in the untreated groups of both categories, particularly in the group that received continuous exposure of PM_2.5_ without treatment. Meanwhile, several areas of parakeratosis together with clear hyperkeratosis were observed in the epidermis in the untreated groups of both experimental categories, as was thickening of the epidermis caused by the accumulation of mainly exudates and a keratin-like substance on the surface of the epidermis. This accumulation may be attributed to inflammation and keratinocyte death caused by PM_2.5_-induced damage. Furthermore, largely enhanced epidermal basal cells were observed in the untreated groups of both categories (Figure 5 and Figure 6), indicating acceleration of the natural regenerative mechanism of the epidermis to replace and repair cells damaged by PM_2.5_ in the upper keratin layer.

Subcutaneous injection of PM_2.5_ also caused moderate injury in the dermis. The findings of both H & E and Masson’s trichrome staining revealed a fragmented, loose, extracellular matrix and loss of extracellular collagen-like substance in the upper dermis as well as focal dermal necrosis. In addition, a decreased number and size of hair follicles and even absence of hair follicles/pilosebaceous units were noted in the dermis; these observations are in accordance with those obtained for the epidermis and confirm the deleterious effects of PM_2.5_ on hair follicles/pilosebaceous units. Moreover, increased and swollen capillaries as well as inflammatory cell and fibroblast infiltration were discovered in the untreated groups of both experimental categories (Figure 5 and Figure 6).

The rats injected with PM_2.5_ and then treated with gel not containing either dosage form of Farn in both categories exhibited features similar to those of the untreated groups, including the presence of numerous follicular cysts in the dermis and epidermis, ruptured cysts, parakeratosis coupled with hyperkeratosis in the epidermis, increased epidermal basal cells, and epithelial loss. In addition, a large decrease in the number and size of hair follicles, a loose extracellular matrix, loss of extracellular collagen-like substance, increased and swollen capillaries, and inflammatory cell and fibroblast infiltration were noted in the dermis (Figure 5 and Figure 6).

An improved arrangement of collagen-like substance and decreased swollen capillaries and inflammatory cell and fibroblast infiltration were noted in the dermis after treatment with 2 mM pure Farn in the group that received continuous injection of PM_2.5_. However, increased follicular cysts, parakeratosis coupled with hyperkeratosis, and a thickened epidermis containing exudates and keratin-like substance were observed in the epidermis, indicating that the protective effect of 2 mM pure Farn on the epidermis was relatively weak. Moreover, 2 mM pure Farn did not exert a restorative effect on either the dermis or epidermis after a 4-week injection of PM_2.5_. These results implicated that 2 mM pure Farn had mild-to-moderate protective effects against the continuous injection of PM_2.5_, whereas 2 mM pure Farn did not show the reparative effects in the post-PM_2.5_ injective experimental category. More well-arranged, collagen-like fragments, fewer swollen capillaries, and no inflammatory cell and fibroblast infiltration were noted in the dermis in the groups treated with 4 mM pure Farn in both experimental categories. However, increased follicular cysts in the epidermis, epidermal exudates, and parakeratosis coupled with hyperkeratosis in some areas were still observed. Wound-healing scores also consistently showed that 4 mM pure Farn exerted mild-to-moderate reparative effects in terms of epithelialization on the epidermis, although 4 mM pure Farn did not exert significant reparative effects on injured hair follicles (Figure 5 and Figure 6).

The groups treated with 2 mM Lipo-Farn in either experimental category exhibited improved arrangement and abundance of the collagen-like substance in the dermis to some extent; however, mild-to-moderate chronic inflammation was still observed in the treated groups. Meanwhile, increased follicular cysts, ruptured cysts, and epithelial loss were noted in the epidermis (Figure 5 and Figure 6). In the epidermis, increased follicular cysts, parakeratosis coupled with hyperkeratosis, increased epidermal basal cells, and epithelial loss were not observed after treatment with 4 mM Lipo-Farn in either experimental category. The collagen-like substance was well-arranged, and PM_2.5_-induced inflammation was completely alleviated. In particular, most of the hair follicle structures were restored in both the experimental groups; this finding was not observed in any other group in either experimental category. These results in 4 mM Lipo-Farn-treated groups in either experimental category suggested that 4 mM Lipo-Farn had the greatest reparative, regenerative, and anti-inflammatory actions on the epidermis, dermis, and hair follicles of the rat skin in both the experimental categories (Figure 5 and Figure 6).

Total wound healing scores further verified that 4 mM Lipo-Farn exhibited the strongest reparative and protective effects on the epidermis, dermis, and whole skin in the groups that received post-PM_2.5_ injection and continuous PM_2.5_ injection. Coherent with our histopathological findings, the wound healing scores also indicated that 4 mM pure Farn as well as 2 mM and 4 mM Lipo-Farn exerted significant regenerative effects on PM_2.5_-induced skin injury (Table 1).

### 2.4. Western Blot Analysis

The protein presence of IL-6 and TNF-α in the rat skin was ascertained by Western blot analysis. The highest IL-6 level was found in the untreated groups in both experimental categories. The Western blot showed that 4 mM Lipo-Farn exhibited the strongest inhibitory effect on IL-6 and TNF-α in both experimental categories, while 2 mM pure Farn/Lipo-Farn and 4 mM pure Farn showed slight-to-moderate suppression on IL-6 level in either category. On the other hand, 2 mM pure Farn and Lipo-Farn and 4 mM pure Farn presented a higher level of TNF-α than those in the untreated groups (Figure 6).

## 3. Discussion

PM_2.5_, a crucial air pollutant, can damage several systems and organs in humans, leading to various diseases. In the current study, we observed that PM_2.5_ induced the production of IL-6 and decreased the viability of skin fibroblasts in vitro. The current in vitro results of PM_2.5_ on skin fibroblasts indicated that treatment with ≥200, ≥400, and ≥800 μg/mL PM_2.5_ disrupted the normal physiological activities of skin fibroblasts, resulted in the induction of inflammatory responses, and decreased the viability of skin fibroblasts, respectively. In addition, the sequential impairment of skin cells in vitro implied the likely deleterious effects of PM_2.5_ on skin in vivo. Subcutaneous injection of PM_2.5_ caused severe impairment in the epidermis and hair follicles as well as moderate injuries in the dermis. Pure Farn or Lipo-Farn in gelatin/HA/xanthan gel was topically applied to treat the adverse effects of the continuous subcutaneous injection of PM_2.5_ on the skin. 

The cell viability-inhibitory data of pure Farn in the current study is similar to that of our previous findings [21]. Furthermore, the cell morphology results revealed that treatment with 0.4 mM pure Farn led to loss of the normal spindle-like shape in approximately 50% of skin fibroblasts in vitro (Figure 3B); this finding is in line with that of our previous study [21]. These results indicated that pure Farn at a concentration of >0.4 mM exerted adverse effects on skin fibroblasts. The MTT assay revealed that liposome encapsulation largely reduced the inhibitory effect of Farn on the viability of skin fibroblasts (Figure 4A). Furthermore, the live–dead cell staining showed that the fluorescence staining of live cells had decreased by approximately 20% after treatment with 1.0 mM Lipo-Farn (Figure 4B). The aforementioned results of the MTT assay and live–dead cell staining are consistent with those of our previous study and indicate that liposome encapsulation could reduce the cytotoxic effects of Farn on skin fibroblasts in vitro. The liposome encapsulation rate for Farn was 69%, and Lipo-Farn was determined to be 342 ± 90 nm in diameter (Figure 1). Although approximately 30% of the Farn was released from Lipo-Farn after incubation for 192 h (Figure 1), our in vivo results indicated that Lipo-Farn could still exert protective and therapeutic effects on PM_2.5_-induced severe damage in the epidermis, hair follicles, and dermis. These findings are supported by those of previous studies reporting that nanoparticles approximately 300 nm in size penetrated deeper into the pilosebaceous unit than did nanoparticle elements [22]. Cyproterone acetate—which is used to treat skin disorders such as acne, hirsutism, and alopecia—accumulated the most in hair follicles when administered in the form of a 300 nm nanostructured lipid carrier [23]. The findings of previous studies and the current study have consistently indicated that nanoparticles approximately 300 nm in size exert great favorable therapeutic effects on skin injuries and disorders in vivo. The histopathological findings demonstrated that PM_2.5_ caused severe injuries to pilosebaceous units in the untreated groups. Loss of the epithelium (epidermal erosion) was observed in the untreated groups of both categories, suggesting that PM_2.5_ exerted highly deleterious effects on the epidermis. The impairment on hair follicles, inflammatory cells, and fibroblast infiltration as well as loose extracellular matrix and loss of extracellular collagen-like substance in the upper dermis indicated moderate damage caused by PM_2.5_ in the dermis. These histopathological findings verified the inflammatory effects of PM_2.5_ on the skin and are consistent with those of previous studies [9,10,12]. Moreover, the groups injected with PM_2.5_ and then treated with HPMC gel not containing either dosage form of Farn in both categories displayed histopathological features similar to those of the untreated groups, indicating that the basal gel did not possess either anti-inflammatory or skin-restorative actions against PM_2.5_ caused impairment on the rat skin.

Our histopathological data showed that treatment with 4 mM Lipo-Farn exerted the strongest reparative, regenerative, and anti-inflammatory effects on the epidermis, dermis, and hair follicles of rat skin in both experimental categories. Treatment with pure 4 mM Farn did not exert a similar reparative effect on injured hair follicles. These findings indicated that liposomal encapsulation of Farn is crucial and beneficial for the repair of hair follicles, likely because it increases the transdermal penetration capacity through pilosebaceous units. That is, 4 mM Lipo-Farn could exert the strongest regenerative and anti-inflammatory effects on hair follicles damaged by PM_2.5_ in the epidermis and dermis in this present study. 

Compared with results in the untreated and 4 mM Lipo-Farn-treated groups, the histopathological findings of the rats received 2 mM Lipo-Farn in both experimental categories suggested that a 2 mM dose of Lipo-Farn was insufficient for the repair of injured hair follicles in the epidermis. Moreover, 2 mM pure Farn did not exert reparative effects on the damage caused by PM_2.5_ in the epidermis and could not alleviate inflammatory responses; a few pustule-like areas were also observed in the groups treated with 2 mM pure Farn. These findings indicated that 2 mM Lipo-Farn or pure Farn could not restore the injured epidermis or alleviate PM_2.5_-induced inflammation, although it enhanced the arrangement and abundance of collagen-like substance in the dermis. 

Our histopathological findings revealed that 2 mM pure Farn exerted mild-to-moderate protective effects against the continuous injection of PM_2.5_; however, the reparative effects of 2 mM pure Farn were not observed in the post-PM_2.5_ injective experimental category, indicating that only higher doses of Farn may exert reparative and regenerative effects on both the dermis and epidermis. The findings in 4 mM pure Farn-treated groups in either experimental category sensibly confirmed this proposal. Our data indicated that 4 mM pure Farn exerted moderate restorative and anti-inflammatory effects on PM_2.5_-induced skin injury in the group that received post-PM_2.5_ injection and the group that received continuous injection of PM_2.5_ despite lack of restorative effects on hair follicles/pilosebaceous units. The significant hair-follicle-restorative effects exerted by 4 mM Lipo-Farn can be attributed to the greater solubility of Lipo-Farn than of pure Farn; higher solubility meant that more Farn could penetrate the skin barrier and be sustained in the hair follicles of the rats. The histopathological findings and wound healing scores in our study indicate that liposome encapsulation is crucial and necessary if Farn is to fully exert its anti-inflammatory, reparative, and regenerative effects on hair follicles in the dermis and epidermis in vivo. 

The strongest inhibitory effect of 4 mM Lipo-Farn on IL-6 and TNF-α in both experimental categories exhibited by Western blot was coherent with the histopathological data and wound-healing scores. Furthermore, the groups treated with 2 mM pure Farn and Lipo-Farn and 4 mM pure Farn displayed a higher level of TNF-α than did the untreated groups, implying that TNF-α plays crucial roles in the regulation of inflammation and wound healing in skin. The TNF-α level in wound tissues was higher than the normal basal level, suggesting that TNF-α may be correlated with the wound-healing process [24]. Moreover, compared with the control mice, the mice treated with the anti-TNF-α monoclonal antibody on day 3 had delayed wound closure, greater distances between the edges of the panniculus carnosus, and decreased inflammatory cell and fibroblast density, again indicating that TNF-α may be essential for would healing in skin [24]. TNF-α was reported to play a crucial positive role in the early process of skin wound healing, demonstrating that TNF-α administration exerted a beneficial effect on this response [24]. In addition to directly regulating inflammatory cytokine production, TNF-α may further regulate the skin’s wound-healing process [25]. Thus, these findings suggested that 4 mM Lipo-Farn not only strongly suppressed proinflammatory responses mediated by TNF-α but also directly promoted epidermal and dermal repair, which was shown in our previous studies [14,20,21]. Because the groups treated with 2 mM pure Farn and Lipo-Farn and 4 mM pure Farn exhibited steady upregulation of TNF-α (Figure 5 and Figure 6), TNF-α could partially exert beneficial effects on wound healing in PM_2.5_-induced skin injury.

The in vitro experiments revealed that PM_2.5_ dose-dependently increased IL-6 production in L929 murine skin fibroblasts. Moreover, the amount of IL-6 detected through Western blot analysis was consistent with histopathologic results demonstrating acute inflammatory responses induced by PM_2.5_ despite the presence of some nonsignificant differences between the groups. The untreated groups in both experimental categories exhibited the highest IL-6 level, whereas only the treatment involving 4 mM Lipo-Farn in both the experimental categories exerted the strongest inhibitory effects on IL-6. Together with our in vitro IL-6 data, these findings indicate that IL-6 mainly plays a proinflammatory role in injured skin. Moreover, IL-6 was demonstrated to be essential in acute inflammation and for the appropriate start of the subsequent wound-healing process [26]. A decreased IL-6 level reflected alleviation of inflammation in the skin. This is in accordance with the previous finding that the IL-6 level was significantly decreased during the remodeling phase in skin’s normal wound-healing process [27]. Moreover, our results support the finding that IL-6 serves as a switch that exerts reparative effects during the inflammatory progress [28]. Compared with the untreated groups and the group treated with 4 mM Lipo-Farn, the group treated with pure Farn and Lipo-Farn had a moderate IL-6 level (Figure 5 and Figure 6); this finding is consistent with the histopathological analysis findings and wound-healing scores obtained in the current study. Moreover, these results are in accordance with those of our previous studies indicating that Farn regulates the IL-6 level in third-degree burns and UVB-caused sunburn in murine skin [14,21].

## 4. Materials and Methods 

### 4.1. Materials 

Farn was purchased from Sigma (St. Louis, MO, USA). HPMC (hydroxypropyl Methylcellulose), HA (hyaluronic acid), cholesterol, lectin, distearoylphosphatidylcholine (DSPC; molecular weight: 790.15 Da), chloroform, and methanol were also procured from Sigma (St. Louis, MO, USA). All reagents used in this study were of reagent grade. 

### 4.2. Production and Characterization of Liposome-Encapsulated Farn

Liposome-encapsulated farnesol (Lipo-Farn) was prepared using the evaporation–sonication method as described in our previous study. Briefly, a mixture of DSPC (8.9 mg), cholesterol (2.7 mg), and Farn (2.5 μL) was used as phospholipids for liposomes. DSPC, cholesterol, lecithin, and Farn were dissolved in methanol:chloroform (1:1, *v*/*v*) and then placed in a flask. The mixture in the flask was homogenized for 2 min (UP 200S, Hielscher, Teltow, Germany) and dried using a rotary evaporator (N-1300, EYELA, Tokyo, Japan) to form a thin film. The produced film was rehydrated with deionized water (5 mL) and sonicated for 5 min (Figure 7A). 

The entrapment efficiency (EE) of Farn in liposomes was determined as follows: 1 mL of the solution containing the prepared Lipo-Farn was centrifuged at 10,000 rpm for 10 min, and the amount of nonencapsulated Farn in the supernatant was measured through high-performance liquid chromatography (HPLC, Agilent 1100 series, Sana Clara, CA, USA). The EE was calculated using the following formula:$$\mathrm{EE}\%=\left(\frac{\mathrm{Total}\;\mathrm{amount}\;\mathrm{of}\;\mathrm{Farn}-\mathrm{Amount}\;\mathrm{of}\;\mathrm{nonencapsulated}\;\mathrm{Farn}}{\mathrm{Total}\;\mathrm{amount}\;\mathrm{of}\;\mathrm{Farn}}\right)\times 100\%$$

The in vitro release of Farn from liposomes was assessed. A total of 1 mL of Lipo-Farn was added to a 1.5 mL microcentrifuge tube. Subsequently, the tube was placed on a shaker with the temperature and shaking rate set at 37 °C and 40 rpm, respectively. At predetermined time points, the sample was centrifuged at 14,000 rpm for 60 min. The amount of nonencapsulated Farn in the supernatant was determined through HPLC. The in vitro release rate of Farn from liposomes was calculated as follows:$$\mathrm{In}\;\mathrm{vitro}\;\mathrm{release}\%=\left(\frac{\mathrm{Total}\;\mathrm{amount}\;\mathrm{of}\;\mathrm{Farn}-\mathrm{Residue}\;\mathrm{of}\;\mathrm{Farn}}{\mathrm{Total}\;\mathrm{amount}\;\mathrm{of}\;\mathrm{Farn}}\right)\times 100\%$$

### 4.3. In Vitro Cell Viability Tests for Lipo-Farn, Farn, and PM_2.5_

L929 mouse fibroblasts were suspended at a density of 7 × 10^3^ cells/mL and seeded onto 96-well plates. Various concentrations of Lipo-Farn, Farn, or PM_2.5_ were added to each well in triplicate. The cell viability of the L929 fibroblasts was examined using the 3-(4,5-dimethylthiazol-2-yl)-2,5-diphenyltetrazolium bromide (MTT) assay. After 24 h exposure to treatment, 20 μL of MTT solution (5 mg/mL) were added to each well, and the cells were incubated for an additional 3 h. The formed formazan precipitate was dissolved in 200 μL of dimethyl sulfoxide, the solution was vigorously mixed to dissolve the dye, and absorbance was measured at 570 nm by using a multiplate reader (Thermo Scientific, Waltham, MA, USA).

Cell viability was examined by performing a live–dead cell assay (Invitrogen, Carlsbad, CA, USA). Briefly, 1 mL of phosphate-buffered saline (PBS) containing 2.5 μL/mL of 4 μM ethidium homodimer-1 (EthD-1) assay solution and 1 μL/mL of 2 μM calcein AM solution was prepared. This assay solution (100 μL) was added to the culture, and the mixture was placed in 37 °C and 5% CO_2_ for 15 min. The sample was observed using a fluorescence microscope at excitation wavelengths 494 nm (green, calcein) and 528 nm (red, EthD-1; Olympus IX71, Tokyo, Japan).

### 4.4. Determination of PM_2.5_-induced Inflammatory Response

L929 fibroblasts (2 × 10^5^ cells/well) were seeded in a 6-well plate and incubated for 24 h. Subsequently, the cells were treated with various PM_2.5_ concentrations (200–1000 μg/mL). After incubation for 24 h, the medium was collected and centrifuged at 2000× *g* for 10 min to obtain a cell-free supernatant. This supernatant was subsequently used in the enzyme-linked immunosorbent assay (ELISA) for IL-6. IL-6 was quantified using the Elabscience mouse IL-6 ELISA kit (Minneapolis, MN, USA), and absorbance was measured at 450 nm following the manufacturer’s protocol. 

### 4.5. In Vivo Animal Experiments 

#### 4.5.1. Establishment of a PM_2.5_-Induced Skin Inflammation Model in Rats

Animal experiments conducted in this study were approved by the Institutional Animal Care and Use Committee of I-Shou University, Taiwan (approval no.: IACUC-ISU107-022, approval date: 28 December 2018). A total of 19 six-week-old Sprague–Dawley female rats (approximately 250 g/each) were used to establish the skin inflammation model. The dorsal skin of the rats was treated with four doses of 100 μL of PM_2.5_ solution (200 μg/mL) per week for 4 weeks. The rats were randomized into the experimental group (with treatment; *n* = 16) and control group (without any treatment; *n* = 3). All the rats were provided access to normal food and water postoperatively. The experimental rats were randomly divided into four experimental groups: Group A (the rats were treated with 2 mM pure Farn gel), Group B (the rats were treated with 4 mM pure Farn gel), Group C (the rats were treated with 2 mM Lipo-Farn gel), and Group D (the rats were treated with 4 mM Lipo-Farn gel). In addition to the untreated control group, the rats in the negative control group received pure Farn gel treatment after the induction of skin inflammation. The gel was prepared by dissolving HPMC (2%) in 90 °C distilled water. The solution was cooled to room temperature, and HA (0.5%) and xanthan gum (0.5%) were added and mixed in through stirring (pure gel). Subsequently, pure Farn (2 or 4 mM) and Lipo-Farn (2 or 4 mM) were added and stirred at 100 rpm for 10 min to prepare Farn gel. 

In vivo animal experiments were performed up to the sixth week (Figure 7B). The first 4-week period was designated for the development of PM_2.5_-induced skin inflammation, followed by treatment with pure Farn gel and gel containing pure Farn or Lipo-Farn (three treatments/week) for 2 weeks. Subsequently, the rats were sacrificed, and their skin samples were harvested to examine skin repair by performing histopathological analysis. To simulate the real effect of PM_2.5_ on skin, we performed another animal study in which a continuous subcutaneous injection of PM_2.5_ was administered during Farn treatment. The subcutaneous injection of PM_2.5_ was administered under anesthesia using Zoletil (tiletamine with zolazepam, 40 mg/kg, intraperitoneal [i.p.], 50 mg/kg, i.p.) and xylazine (10 mg/kg, i.p.). During the in vivo animal study period, clinical signs of pain, salivation, and abnormal behavior were carefully monitored. 

#### 4.5.2. Histopathological Analysis

After a 2-week treatment with pure Farn, Lipo-Farn, or control, the skin of the rats was harvested and fixed in 10% neutral-buffered formalin. The skin samples were then dehydrated in graded ethanol solutions, cleared in xylene, embedded in paraffin blocks, and cut into 5 μm thick sections. Hematoxylin and eosin (H & E) staining was performed for histopathological examination of the skins. Furthermore, Masson trichrome staining was conducted to assess changes in the collagen content of the skin of the rats treated with Farn or Lipo-Farn. ImageJ software (Version 1.50; National Institutes of Health, Bethesda, MD, USA) was used to measure the collagen content in each group. The color settings employed in ImageJ remained constant throughout the analysis of Masson’s trichrome-stained areas in every sample. The samples were evaluated at 100× magnification, and the calculation was repeated in three microscopic fields.

#### 4.5.3. Western Blot Analysis

To examine the inflammation-alleviating and protective effects of Farn or Lipo-Farn on the rat skin treated with PM_2.5_, the rat skin specimens were harvested 2 weeks after treatment. The samples were placed in precooled (4 °C) PBS solution. The samples were then ground until fully homogenized and added to ice-cold lysis buffer; the tissue homogenate was lysed on ice for 0.5–1 h. After 1 h, the tissue homogenate was centrifuged at 13,000 rpm at 4 °C for 15 min. The proteins were separated through 10% sodium dodecyl sulfate–polyacrylamide gel electrophoresis and subsequently transferred and blotted onto a nitrocellulose membrane. Nonspecific binding sites on the membrane were blocked using 5% skim milk powder in Tris buffered saline containing 0.05% Tween 20 for 1 h at room temperature. The membrane was then incubated with primary antimouse IL-6, TNF-α, and β-actin antibodies (Santa Cruz, CA, USA) overnight at 4 °C. The ratio of the IL-6 and TNF-α protein levels in the experimental groups to those in the control group was measured through semiquantitative intensity analysis (normalized by the respective β-actin and background) by using ImageJ.

### 4.6. Statistical Analysis

All values are expressed as the mean ± standard error of the mean. One-way analysis of variance was used to identify significant differences between the experimental and control groups. A *p* value < 0.05 was considered statistically significant. All statistical analyses were performed using SPSS version 20.0.

## 5. Conclusions

Taken together, the results of this study revealed that PM_2.5_ caused cytotoxicity and induced IL-6 production in skin fibroblasts in vitro. Subcutaneous injection of 200 μg/mL PM_2.5_ further caused severe-to-moderate injury of hair follicles, the epidermis, and the dermis. In addition, acute and chronic inflammatory responses were observed in the groups receiving post-PM_2.5_ and continuous subcutaneous injection of PM_2.5_. Liposome encapsulation largely reduced the cytotoxicity of Farn to skin fibroblasts. In both the post- and continuous PM_2.5_-treated experimental groups, 4 mM Lipo-Farn largely restored PM_2.5_-induced damage in the epidermis and dermis, regenerated injured hair follicles, and alleviated acute and chronic inflammation induced by PM_2.5_ in the skin, suggesting that the reduction of cytotoxicity and increase in skin permeability achieved through liposome encapsulation could increase and reinforce the beneficial effects of Farn on the skin. The findings of this study validate the therapeutic and protective effects of Lipo-Farn against the diverse impairment caused by PM_2.5_ in skin. 

## Figures and Tables

**Figure 1 ijms-22-06076-f001:**
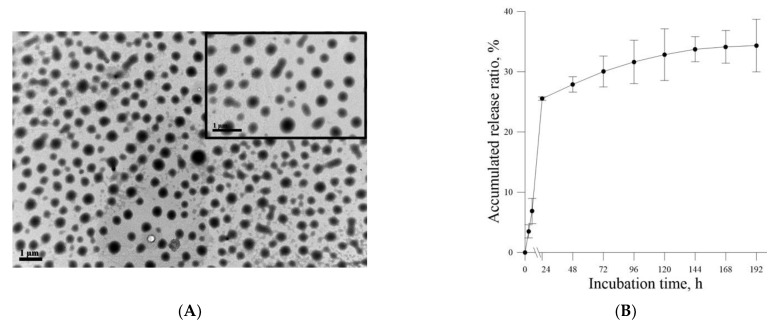
(**A**) Transmission electron microscopy images of Lipo-Farn. The diameter of the liposomes was 342 ± 90 nm. (**B**) Farn release profile from Lipo-Farn. The release rate of Farn was 26% after 24 h of incubation.

**Figure 2 ijms-22-06076-f002:**
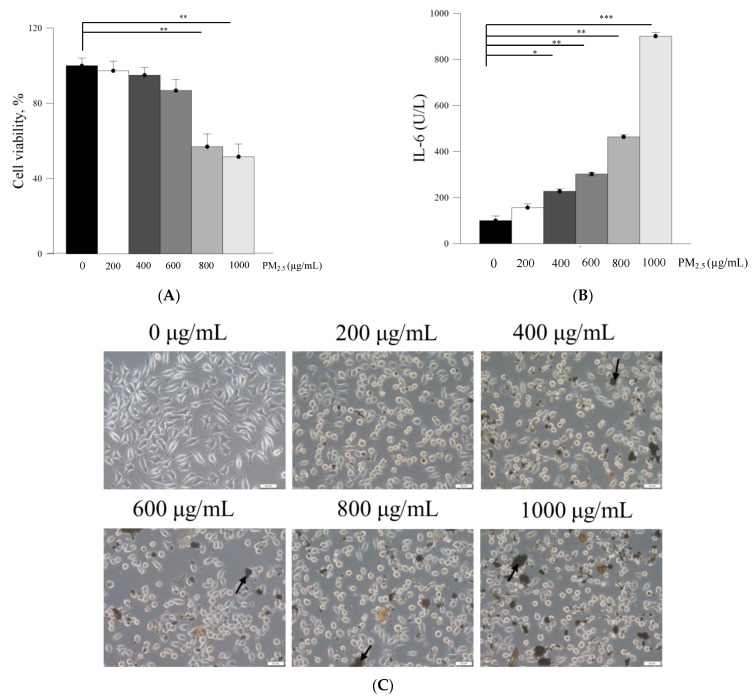
(**A**) 3-(4,5-Dimethylthiazol-2-yl)-2,5-diphenyltetrazolium bromide (MTT) cell viability assay of PM_2.5_ for L929 fibroblasts. (**B**) Interleukin-6 (IL-6) level in L929 fibroblasts induced after treatment with PM_2.5_ as measured using an enzyme-linked immunosorbent assay. (**C**) Cell morphology after treatment with different PM_2.5_ concentrations. Black arrow indicated the precipitated and aggregated PM_2.5_ particles. * *p* < 0.05, ** *p* < 0.01, *** *p* < 0.001 by analysis of variance. Bar: 50 μm.

**Figure 3 ijms-22-06076-f003:**
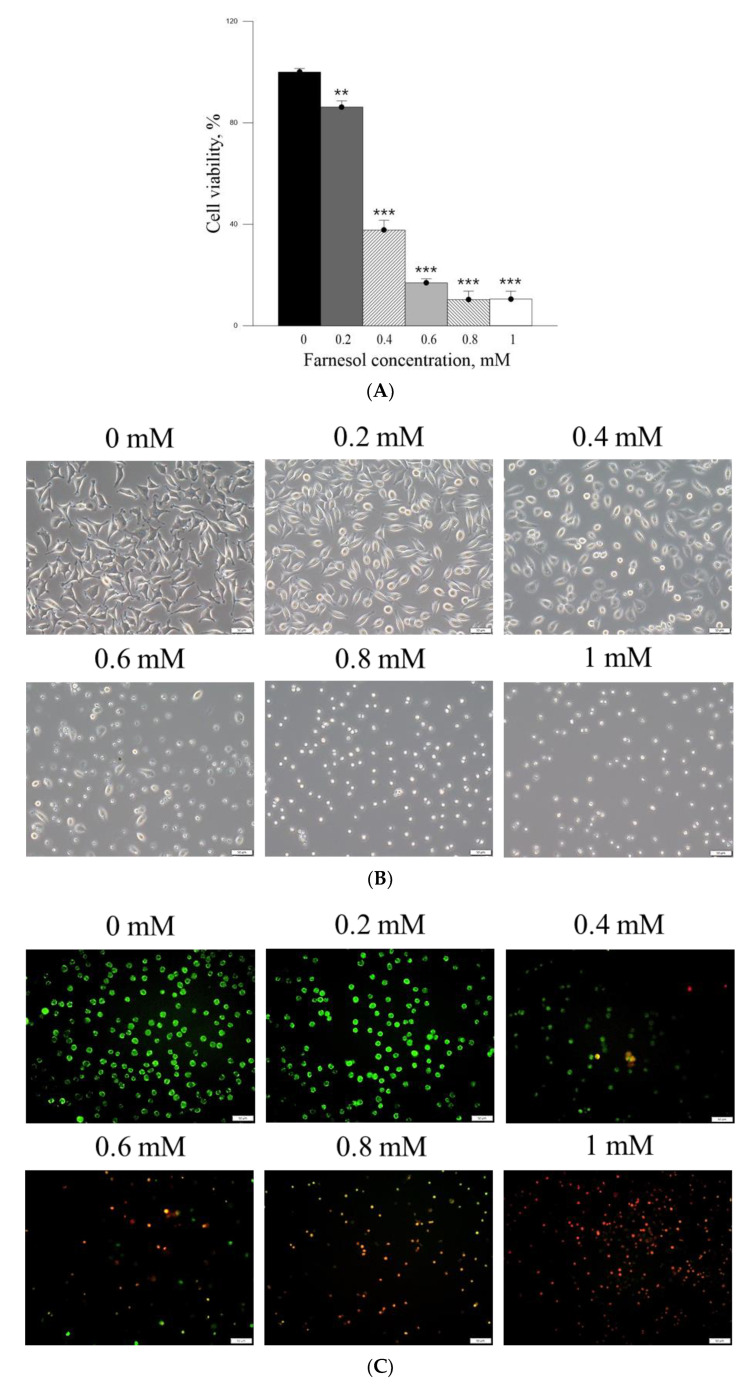
(**A**) MTT cell viability assay of pure Farn in L929 fibroblasts. (**B**) Cell morphology of L929 fibroblasts treated with different concentrations of pure Farn. (**C**) Live–dead staining of cells treated with pure Farn at different concentrations (green fluorescence, live cells; red fluorescence, dead cells). ** *p* < 0.01, *** *p* < 0.001, by analysis of variance compared with the control group (0 mM pure Farn). Bar: 50 μm.

**Figure 4 ijms-22-06076-f004:**
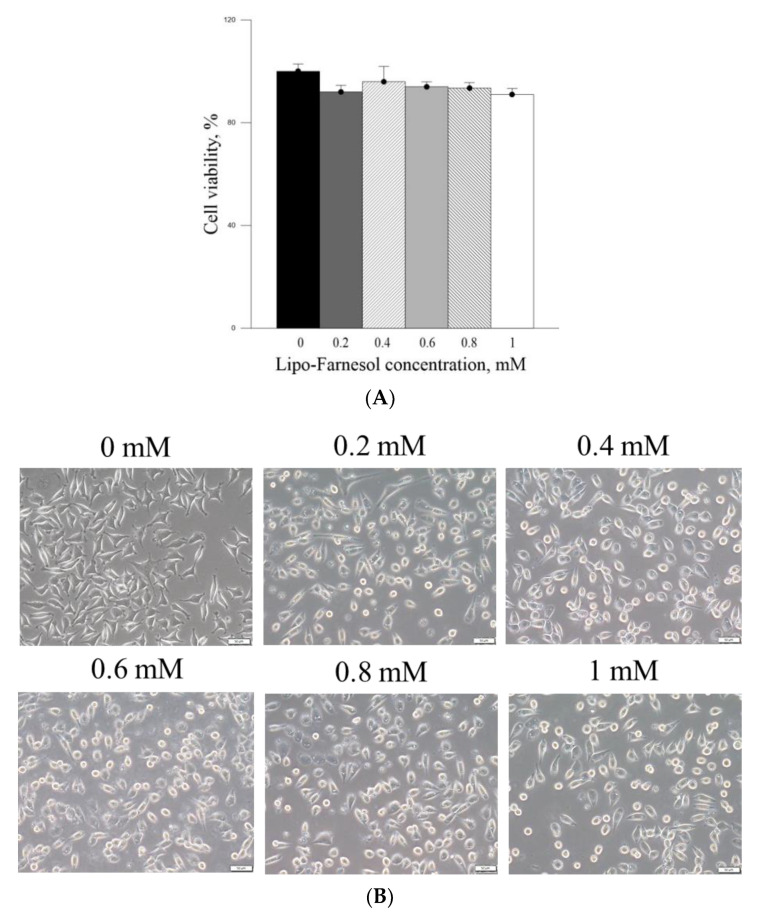
(**A**) MTT cell viability assay of Lipo-Farn in L929 fibroblasts. (**B**) Cell morphology of L929 fibroblasts treated with Lipo-Farn. (**C**) Live–dead staining of L929 fibroblasts treated with different concentrations of Lipo-Farn (green fluorescence, live cells; red fluorescence, dead cells). Bar: 50 μm.

**Figure 5 ijms-22-06076-f005:**
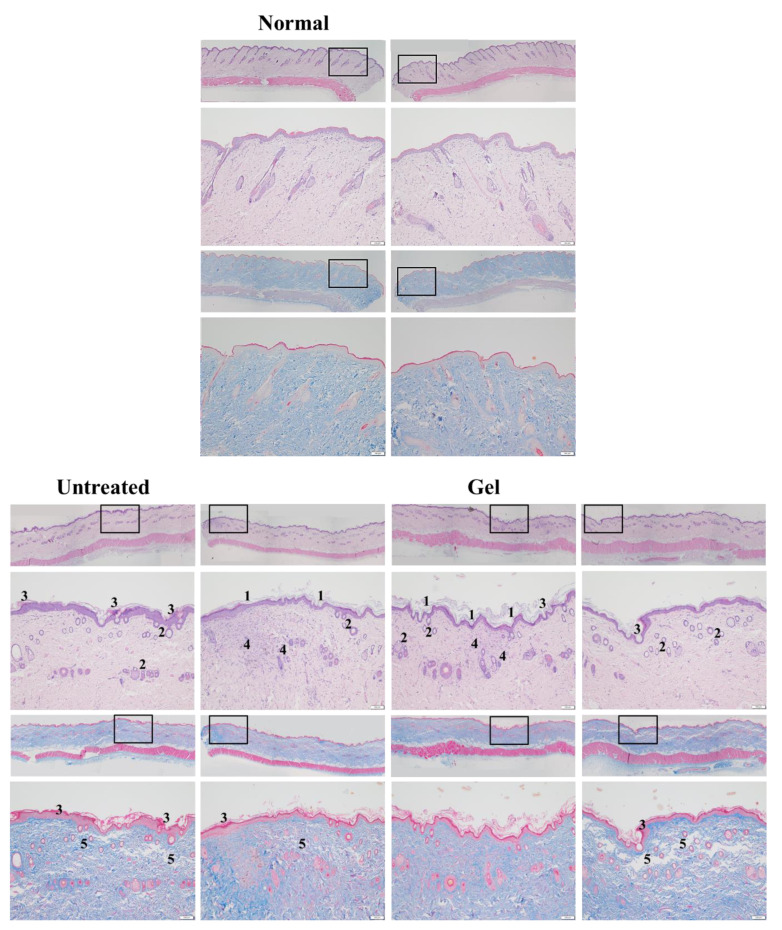
(**A**) Hematoxylin and eosin (H & E) staining and Masson’s trichrome staining for rat normal skin and the groups in post-PM_2.5_ subcutaneous injective treatment category. In the histopathological analysis, different lesions caused by PM_2.5_ were observed as follows. 1. Epidermal erosion, loss of keratin, parakeratosis coupled with hyperkeratosis. 2. Epithelial and follicle cysts arising from injured pilosebaceous units. 3. Rupture of the cyst lining, accumulation of exudates and keratin-like substance in the epidermis. 4. Inflammatory cell and fibroblast infiltration. 5. Fragmented, loose extracellular matrix and loss of extracellular collagen-like substance. Bar: 100 μm. (**B**) Western blot analysis of IL-6 and TNF-α. The rat skin specimens were collected from the groups treated with or without gel, Farn, and Lipo-Farn after the subcutaneous injection of PM_2.5_ (0.4 mL, 200 μg/mL).

**Figure 6 ijms-22-06076-f006:**
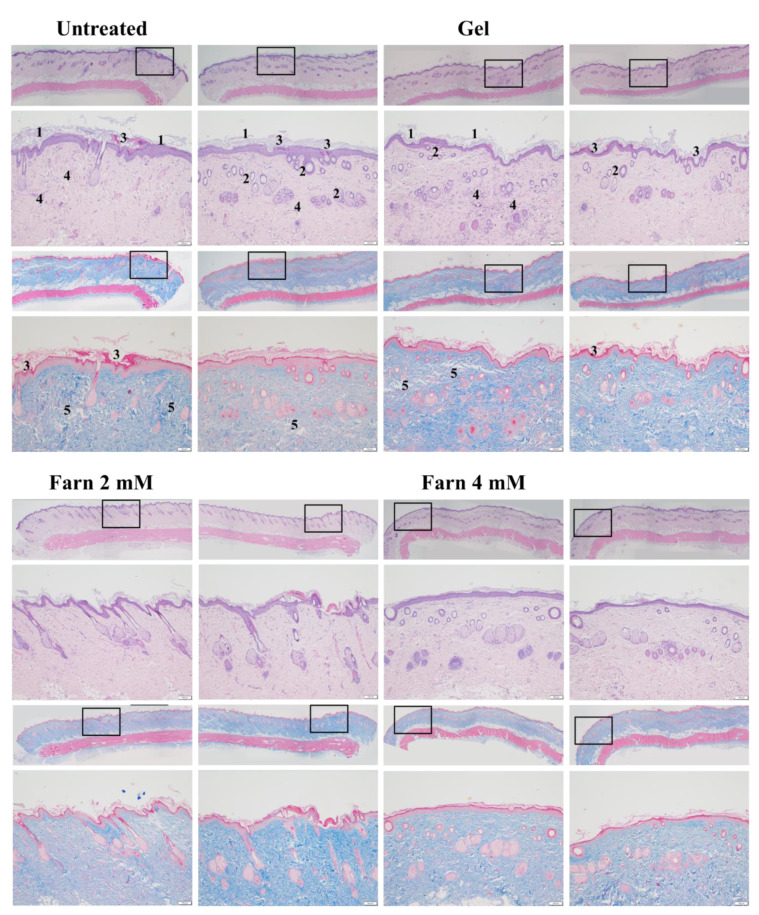
(**A**) H & E and Masson’s trichrome staining for the groups in continuous PM_2.5_ subcutaneous injection and simultaneous treatment category. In the histopathological analysis, different lesions caused by PM_2.5_ were observed as follows. 1. Epidermal erosion, loss of keratin, parakeratosis coupled with hyperkeratosis. 2. Epithelial and follicle cysts arising from injured pilosebaceous units. 3. Rupture of the cyst lining, accumulation of exudates and keratin-like substance in the epidermis. 4. Inflammatory cell and fibroblast infiltration. 5. Fragmented, loose extracellular matrix and loss of extracellular collagen-like substance. Bar: 100 μm. (**B**) Western blot analysis of IL-6 and TNF-α. The rat skin specimens were collected from the groups treated with or without gel, Farn, and Lipo-Farn after the subcutaneous injection of PM_2.5_. (Continuous PM_2.5_ subcutaneous injection and simultaneous treatment category).

**Figure 7 ijms-22-06076-f007:**
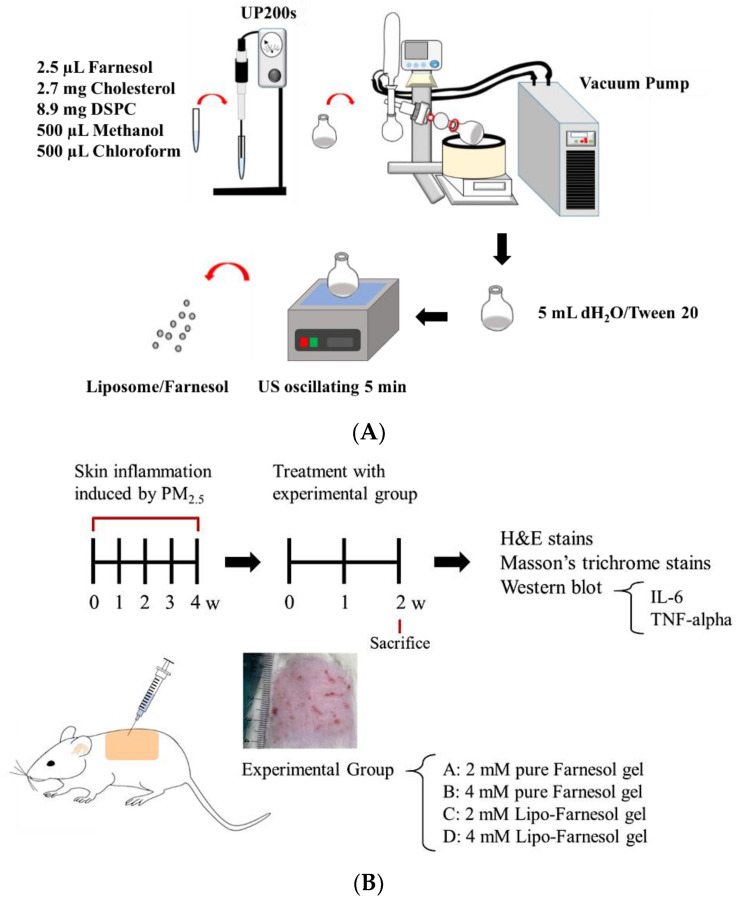
(**A**) Schematic of the process of liposome encapsulation of farnesol (Farn). (**B**) Timescale of post-PM_2.5_ subcutaneous injective treatment with pure Farn or liposome-encapsulated Farn (Lipo-Farn; Category 1) and the continuous subcutaneous injection of PM_2.5_ coupled with simultaneous Farn or Lipo-Farn treatment (Category 2). PM_2.5_: particulate matter with aerodynamic diameter ≤2.5 μm.

**Table 1 ijms-22-06076-t001:** Wound healing scores in the epidermis and dermis in the rats received Farn, Lipo-Farn, gel or none in post-PM_2.5_ subcutaneous injection and continuous PM_2.5_ subcutaneous injection categories.

1. Post-PM2.5 Subcutaneous Injective Treatment Category 1						
	**A**	**B**	**C**	**D**	**B**	**F**
Epithelialization	1.00 ± 0.00	1.00 ± 0.00	1.33 ± 0.33	2.33 ± 0.33 *^,#^	2.00 ± 0.00	3.33 ± 0.33 ***^,###^
Regeneration and reparation of hair follicle	1.00 ± 0.00	1.33 ± 0.33	1.67 ± 0.33	1.67 ± 0.33	2.33 ± 0.33	3.67 ± 0.33 ***^,###^
Acute inflammation	2.00 ± 0.00	2.67 ± 0.33	2.00 ± 0.00	2.67 ± 0.33	2.67 ± 0.33	3.67 ± 0.33 **
Subchronic to chronic inflammation	0.33 ± 0.33	0.67 ± 0.33	0.67 ± 0.33	3.67 ± 0.33 ***^,###^	1.67 ± 0.33	4.00 ± 0.00 ***^,###^
Collagenization	1.33 ± 0.33	1.67 ± 0.33	2.00 ± 0.33	3.33 ± 0.33 **^,#^	2.33 ± 0.33	3.67 ± 0.33 **^,##^
Total	5.67 ± 0.33	7.33 ± 0.67	7.67 ± 0.88	13.67 ± 0.88 ***^,###^	11 ± 0.58 ***^,#^	18.33 ± 0.33 ***^,###^
**2. Continuous PM_2.5_ Subcutaneous Injection and Simultaneous Treatment Category 2**						
	**A**	**B**	**C**	**B**	**E**	**F**
Epithelialization	0.33 ± 0.33	0.33 ± 0.33	0.67 ± 0.33	2.00 ± 0.00 **^,##^	2.33 ± 0.33 **^,##^	3.00 ± 0.00 ***^,###^
Regeneration and reparation of hair follicle	0.67 ± 0.33	0.67 ± 0.33	1.00 ± 0.00	2.33 ± 0.33 *^,#^	1.67 ± 0.33	3.67 ± 0.33 ***^,###^
Acute inflammation	1.67 ± 0.33	1.67 ± 0.33	2.33 ± 0.33	3.00 ± 0.00 *^,#^	2.67 ± 0.33	3.00 ± 0.00 *^,#^
Subchronic to chronic inflammation	1.67 ± 0.33	1.33 ± 0.33	3.00 ± 0.00 *^,##^	2.67 ± 0.33 ^#^	3.33 ± 0.33 **^,##^	3.00 ± 0.00 *^,##^
Collagenization	1.33 ± 0.33	1.67 ± 0.33	2.33 ± 0.33	2.67 ± 0.33 *	2.67 ± 0.33 *	3.67 ± 0.33 ***^,##^
Total	5.67 ± 0.67	5.67 ± 0.67	9.33 ± 0.67 *^,#^	12.67 ± 0.33 ***^,##^	12.67 ± 0.67 ***^,###^	16.33 ± 0.67 ***^,###^

Acute/subchronic/chronic inflammation: severe (0) moderate (1) mild (2) mild-none (3) none (4). The scores were expressed as mean ± SEM, significant differences were determined by one-way ANOVA with Tukey–Kramer test. * *p* < 0.05, ** *p* < 0.01, *** *p* < 0.001, compared to untreated group. # *p* < 0.05, ## *p* < 0.01, ### *p* < 0.001, compared to gel-treated group. Group A: Untreated group (negative control 1). Group B: Treated with Gel (negative control 2). Group C: Treated with 2 mM pure Farn. Group D: Treated with 4 mM pure Farn. Group E: Treated with 2 mM Lipo-Farn. Group F: Treated with 4 mM Lipo-Farn.

## Data Availability

The data presented in this study are available in the paper.

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
