# Peer review of "Farnesol-Loaded Liposomes Protect the Epidermis and Dermis from PM_2.5_-Induced Cutaneous Injury"

_ijms, 2021, doi:10.3390/ijms22116076_

Round 1
Reviewer 1 Report
This manuscript describes the effect of particulate matter with aerodynamic diameter ≤ 2.5 μm (PM2.5) on oxidative stress through free radical generation and incomplete volatilization.This study examined the deleterious effects of PM2.5 on the epidermis and dermis. In addition, Farn-encapsulated liposomes (Lipo-Farn) and gelatin/HA/xanthan gel containing Lipo-Farn were prepared and applied in vivo. The findings of the current study indicate the therapeutic and protective effects of Lipo-Farn against various injuries caused by PM2.5 in the pilosebaceous units, epidermis, and dermis of skin.
Your work is very interesting, well structured, with reliable results and supported by an appropriate method.
Are the animals involved in the study the same sex and have similar weights and gender? This should be clarified in methods.
What are the black dots in images in Fig. 3?
Something is wrong with the numbers in Fig. 3. Please correct it.
Fig. 7: a part of the blot is covered by the graph. Please correct it.
Author Response
The revised manuscript has been checked by the Wallace Academic Editing and an English-spoked associate professor (Ioannis Manousakas)
Reviewer 1 comments:
- Are the animals involved in the study the same sex and have similar weights and gender? This should be clarified in methods.
Response: We greatly appreciate the reviewer’s comment. We have provided the information of rat in the methods (A total of 19 six-week Sprague–Dawley female rats (approximately 250 g) were used to establish the skin inflammation model).
- What are the black dots in images in Fig. 3?
Response: We greatly appreciate the reviewer’s comment. When the concentration of PM2.5increased, PM2.5 particles precipitated and aggregated to form black dots in culture medium (as shown in Fig). We have marked these black dots by black arrow.
- Something is wrong with the numbers in Fig. 3. Please correct it.
Response: We greatly appreciate the reviewer’s comment. We have corrected the sequences of (A), (B) and (C) and their legends.
- Fig. 7: a part of the blot is covered by the graph. Please correct it.
Response: We greatly appreciate the reviewer’s comment. We have corrected the Figure 7.
Reviewer 2 Report
Dear Dr. Kuo,
The manuscript entitled “Farnesol-loaded liposomes protect the epidermis and dermis from PM2.5-induced cutaneous injury” addresses the negative effects of exposure to particulate matter with aerodynamic diameter below 2.5 µm on the skin. Authors have investigated the anti-inflammatory and therapeutic effects of Farnesol formulated into a conventional gel and a liposome-based gel, revealing good tolerability of the nanostructured formulation and positive results in the healing process.
The manuscript is precise and concise, following a logical flow of reading. The study has been well planned and designed.
I have some suggestions,
Format-related issues:
- Review the IJMC authors instructions or the IJMS template, which indicates the manuscript’s structure to be: (1) Introduction; (2) Results; (3) Discussion; (4) Material and methods; (5) conclusions.
Please re-format the manuscript according to the IJMS instructions.
- It would be interesting to give a little more information about Farnesol, the introduction gives enough background, but there are only one or two sentences referring to the active of interest.
- In Figure 2, the caption considers panel A and panel B, please include the letters on the image.
- Figures 6 and 7 are very large; so, although it is clear what image corresponds to each test described in the captions, labelling each panel [(a), (b), (c)] would be useful and more reader-friendly.
In Figure 7., the images of the Western blot are not shown together, some of them overlap.
- Including arrows in the histopathological images would aid the non-expert reader to visualize the cell infiltrations and changes caused by the PM2.5 exposure.
- In section 2.2. the equation for the EE% should be labelled according to the authors' instructions, see IJMS template
- In the Results and Discussion section, paragraph 7 (page 6), the citation format “(Verma et al., 2016)” should be [22].
- And finally, the conclusions section is missing, or the last paragraph of the Results and Discussion section should be moved to the conclusion section.
General:
- Section 2.1. the reagents used for the gel are not listed in the material section (HPMC, HA and xanthan gum). Please, include them.
- Section 2.2. “The produced film was rehydrated with deionized water (5 mL) and sonicated for 5 min (Fig 1A)”. But, figure 1A states 20 minutes, please revise and correct if applicable.
- Section 2.6.1. in the description of the gel preparation, please indicate what the abbreviations HPMC and HA stand for.
- In Results and Discussion section, the second-to-last sentence
“Subcutaneous injection of PM2.5 caused severe impairment in the dermis and hair follicles as well as moderate injuries in the dermis.”
The dermis is repeated, might it be one of them epidermis instead?
- In the Results and Discussion section, paragraph 7 (page 6), authors state that “[…] the amount of Farn that reached injured hair follicles was higher in the groups treated with Lipo-Farn than in those treated with pure Farn in vivo […]”
How was the amount of Farn measured? Do you have any supplementary data?
6. For the in vivo experiments, the Ethics Committee information is missing, please provide the protocol code and the date of approval.
Sincerely,
Author Response
The revised manuscript has been checked by the Wallace Academic Editing and an English-spoked associate professor (Ioannis Manousakas)
Reviewer 2 comments:
The manuscript entitled “Farnesol-loaded liposomes protect the epidermis and dermis from PM2.5-induced cutaneous injury” addresses the negative effects of exposure to particulate matter with aerodynamic diameter below 2.5 µm on the skin. Authors have investigated the anti-inflammatory and therapeutic effects of Farnesol formulated into a conventional gel and a liposome-based gel, revealing good tolerability of the nanostructured formulation and positive results in the healing process.
The manuscript is precise and concise, following a logical flow of reading. The study has been well planned and designed.
I have some suggestions,
Format-related issues:
- Review the IJMC authors instructions or the IJMS template, which indicates the manuscript’s structure to be: (1) Introduction; (2) Results; (3) Discussion; (4) Material and methods; (5) conclusions.
Please re-format the manuscript according to the IJMS instructions.
Response: We greatly appreciate the reviewer’s comment. We have re-formatted the manuscript to the journal paper’s structure.
- It would be interesting to give a little more information about Farnesol, the introduction gives enough background, but there are only one or two sentences referring to the active of interest.
Response: We greatly appreciate the reviewer’s comment. We have added more information about farnesol: The farnesol has been found to promote wound healing by suppressing the production of proinflammatory cytokines such as interleukin (IL)-1 beta, IL-6, and tumor necrosis factor (TNF)-alpha [14, 20, 21]. The present study investigated the deleterious effects of PM2.5 on the epidermis and dermis and examined the anti-inflammatory and restoration-promoting effects of pure and liposomal Farn on various skin injuries caused by PM2.5.
- In Figure 2, the caption considers panel A and panel B, please include the letters on the image.
Response: We greatly appreciate the reviewer’s comment. The captions A and B have added.
- Figures 6 and 7 are very large; so, although it is clear what image corresponds to each test described in the captions, labelling each panel [(a), (b), (c)] would be useful and more reader-friendly.
Response: We greatly appreciate the reviewer’s comment. The captions A and B have added. And we have labelled and described the lesions caused by PM2.5 on the untreated and Gel groups in figure legend: In the histopathological analysis, different lesions caused by PM2.5 were observed as below: 1. epidermal erosion, loss of keratin, parakeratosis coupled with hyperkeratosis. 2. epithelial and follicle cysts arising from injured pilosebaceous units. 3. rupture of the cyst lining, accumulation of exudates and keratin-like substance in the epidermis. 4. inflammatory cell and fibroblast infiltration. 5. fragmented, loose extracellular matrix and loss of extracellular collagen-like substance.
These lesion descriptions could provide more information for the readers.
- In Figure 7., the images of the Western blot are not shown together, some of them overlap.
Response: We greatly appreciate the reviewer’s comment. We have re-arranged the figures.
- Including arrows in the histopathological images would aid the non-expert reader to visualize the cell infiltrations and changes caused by the PM2.5 exposure.
Response: We greatly appreciate the reviewer’s comment.
We have labelled and described the lesions caused by PM2.5 on the untreated and Gel groups images: In the histopathological analysis, different lesions caused by PM2.5 were observed as below: 1. epidermal erosion, loss of keratin, parakeratosis coupled with hyperkeratosis. 2. epithelial and follicle cysts arising from injured pilosebaceous units. 3. rupture of the cyst lining, accumulation of exudates and keratin-like substance in the epidermis. 4. inflammatory cell and fibroblast infiltration. 5. fragmented, loose extracellular matrix and loss of extracellular collagen-like substance.
- In section 2.2. the equation for the EE% should be labelled according to the authors' instructions, see IJMS template
Response: We greatly appreciate the reviewer’s comment. We have corrected the equation format according to the journal request.
- In the Results and Discussion section, paragraph 7 (page 6), the citation format “(Verma et al., 2016)” should be [22].
Response: We greatly appreciate the reviewer’s comment. We have corrected the reference
- And finally, the conclusions section is missing, or the last paragraph of the Results and Discussion section should be moved to the conclusion section.
Response: We greatly appreciate the reviewer’s comment. We have corrected the conclusion accordingly.
General:
- Section 2.1. the reagents used for the gel are not listed in the material section (HPMC, HA and xanthan gum). Please, include them.
Response: We greatly appreciate the reviewer’s comment. We have listed the full names of materials used in this study.
- Section 2.2. “The produced film was rehydrated with deionized water (5 mL) and sonicated for 5 min(Fig 1A)”. But, figure 1A states 20 minutes, please revise and correct if applicable.
Response: We greatly appreciate the reviewer’s comment. We have corrected the figure 1A to 5 min.
- Section 2.6.1. in the description of the gel preparation, please indicate what the abbreviations HPMC and HA stand for.
Response: We greatly appreciate the reviewer’s comment. We have added the full names of materials used in this study
- In Results and Discussion section, the second-to-last sentence
“Subcutaneous injection of PM2.5 caused severe impairment in the dermis and hair follicles as well as moderate injuries in the dermis.”
The dermis is repeated, might it be one of them epidermis instead?
Response: We greatly appreciate the reviewer’s comment. The sentence was corrected to: Subcutaneous injection of PM2.5 caused severe impairment in the epidermis and hair follicles as well as moderate injuries in the dermis.
- In the Results and Discussion section, paragraph 7 (page 6), authors state that “[…] the amount of Farn that reached injured hair follicles was higher in the groups treated with Lipo-Farn than in those treated with pure Farn in vivo […]”
How was the amount of Farn measured? Do you have any supplementary data?
Response: We greatly appreciate the reviewer’s comment. Actually, we did not measure the amount of farnesol that reached hair follicles. We deleted this sentence and changed into conservative descriptions based on the liposome encapsulated nanoparticle could effectively penetrate the skin barrier and from the obtained results: Treatment with pure 4 mM Farn did not exert a similar reparative effect on injured hair follicles. These findings indicated that liposomal encapsulation of Farn is crucial and beneficial for the repair of hair follicles, probably because it increases the transdermal penetration capacity through pilosebaceous units. That is, 4-mm Lipo-Farn could exert the strongest regenerative and anti-inflammatory effects on hair follicles damaged by PM2.5 in the epidermis and dermis in this present study.
- For the in vivo experiments, the Ethics Committee information is missing, please provide the protocol code and the date of approval.
Response: We greatly appreciate the reviewer’s comment. We have provided the whole information of Ethics Committee information in the revised manuscript.
Round 2
Reviewer 1 Report
The authors improved the quality of their manuscript.